# Understanding the impacts of health information systems on patient flow management: A systematic review across several decades of research

**Quy Nguyen**[1]*, **Michael Wybrow**[1], **Frada Burstein**[1], **David Taylor**[2], **Joanne Enticott**[3]

**1** Department of Human-Centred Computing, Faculty of Information Technology, Monash University, Melbourne, Australia, **2** Office of Research and Ethics, Eastern Health, Melbourne, Australia, **3** Monash Centre for Health Research and Implementation, Monash University, Melbourne, Australia

* quy.nguyen@monash.edu

## Abstract

### Background

Patient flow describes the progression of patients along a pathway of care such as the journey from hospital inpatient admission to discharge. Poor patient flow has detrimental effects on health outcomes, patient satisfaction and hospital revenue. There has been an increasing adoption of health information systems (HISs) in various healthcare settings to address patient flow issues, yet there remains limited evidence of their overall impacts.

### Objective

To systematically review evidence on the impacts of HISs on patient flow management including what HISs have been used, their application scope, features, and what aspects of patient flow are affected by the HIS adoption.

### Methods

A systematic search for English-language, peer-review literature indexed in MEDLINE and EMBASE, CINAHL, INSPEC, and ACM Digital Library from the earliest date available to February 2022 was conducted. Two authors independently scanned the search results for eligible publications, and reporting followed the PRISMA guidelines. Eligibility criteria included studies that reported impacts of HIS on patient flow outcomes. Information on the study design, type of HIS, key features and impacts was extracted and analysed using an analytical framework which was based on domain-expert opinions and literature review.

### Results

Overall, 5996 titles were identified, with 44 eligible studies, across 17 types of HIS. 22 studies (50%) focused on patient flow in the department level such as emergency department while 18 studies (41%) focused on hospital-wide level and four studies (9%) investigated network-wide HIS. Process outcomes with time-related measures such as 'length of stay'

**Data Availability Statement:** All relevant data is provided in the article and in the supporting file S3. Copies of the included studies are freely available online.

**Funding:** QDN was supported by a Ph.D. scholarship jointly funded by the Monash University Graduate Research Industry Partnership (GRIP) program and by Eastern Health. The Funders of this work did not have any direct role in the design of the study, its execution, analyses, interpretation of the data, or decision to submit results for publication.

**Competing interests:** No authors have competing interests

and 'waiting time' were investigated in most of the studies. In addition, HISs were found to address flow problems by identifying blockages, streamlining care processes and improving care coordination.

## Conclusion

HIS affected various aspects of patient flow at different levels of care; however, how and why they delivered the impacts require further research.

## 1. Introduction

Patient flow refers to the progressive movement of patients through different units or departments of the care setting. The aim of patient flow management is to provide safe and efficient patient care while assuring the best use of resources [1]. Hospitals around the world have undertaken several efforts and strategies to tackle patient flow problems and to provide high-quality care at the right time and right place. Meanwhile, there is an extensive stream of research reporting methods and interventions addressing patient flow problems. A recent umbrella review [2] found that over 25 different interventions have been used by hospitals around the world to solve the overcrowding issues in the emergency department (ED). However, previous studies focused primarily on interventions for a single, isolated hospital unit or ward with ED being the most frequently mentioned [3, 4]. While many systematic reviews related to patient flow interventions have been done, a summary of these systematic reviews shows that most of these reviews have focused on traditional, non-IT interventions such as triage, streaming, and fast track. Systematic reviews on using health information systems (HISs) to tackle patient flow problems exist; however, they are often limited to a single specific system, such as computer provider order entry (CPOE) system [5]; methods such as computer simulation modelling [6]; or measures such as length of stay (LOS) [7].

HISs have been adopted by health providers to improve patient flow in various healthcare settings. For example, in emergency care, the automatic push notification system was used to address ED congestion, reduce LOS, and decrease patient load by providing updated information and improving patient navigation [8]. Dashboard systems were adopted to coordinate ambulance services and improve access to emergency services across multiple hospitals [9]. HISs provides data about ED visits which were used to create a robust prediction about hospital admissions and increase logistical efficiency [10]. In addition, Blaya et al. [11] investigated the use of HISs in improving access to laboratory results and the quality of care. These are a few examples illustrating the impacts of HISs on patient flow management.

In recent patient flow research, it has been suggested that utilising advanced data analytics techniques for patient flow management can be achieved by adopting HISs. For example, Rutherford et al. [12] claim that data analytics is essential in achieving improvement in systematic-wide flows through its capabilities in matching patient demand and hospital supply. Real-time demand capacity has been successfully implemented in many healthcare organisations to predict and match supply and demand [13]. Similarly, Berg et al. [14] called for a shift in the research paradigm from predicting and controlling to analysing and managing to achieve better flow outcomes. This can be done through the application of information technology in analysing data to proactively manage patient flows. Despite the rich tradition of inquiry in research about the use of HISs in patient flow management, to date, to the best of our knowledge, no systematic review has been conducted to assess the impacts of a broad range of HISs on patient flow management, highlighting an evidence gap in the literature. Therefore, a

systematic review of this topic will provide more complete insights as to how HISs have been adopted for and impacted patient flow management practice.

## 2. Objectives

This systematic literature review aimed to examine and summarise information from published studies on the use of HISs in healthcare settings to manage and improve patient flows. We are interested in exploring what information systems have been adopted for managing patient flow and solving flow problems such as blockages, delays, and overcrowding, and their effectiveness. We examined studies that focused on department-level (e.g., ED), hospital-wide, and network-wide interventions. Particularly, our objectives are to provide critical analysis on:

- Study characteristics: Chronological and geographical distribution of the studies, study settings, and research designs.

- Study contents: What types and features of information system have been used for patient flow management, their results and effectiveness on patient flow outcomes.

## 3. Research questions

This review addresses the following research questions:

- What HISs have been used for hospitals' patient flow management?

- What are the impacts of HISs on patient flow outcomes?

- In what ways, have HISs been used to manage patient flow?

## 4. Method

### 4.1 Search strategy

We searched for peer-reviewed journal articles published in English from MEDLINE and EMBASE via Ovid, CINAHL, INSPEC, and ACM Digital Library from the earliest date available to February 2022. In addition, we examined the reference lists of the search results to retrieve further eligible papers. The search was conducted from June 2020 to July 2020 and then re-run in February 2022 before the data extraction process.

  With the assistance of a subject librarian, we developed a systematic search strategy for this review (S1 File). To obtain the most comprehensive search results, we employed medical subject headings (MeSH) keywords when they are available in combination with free text keywords from the PICOS framework. We combined the following terms (Table 1) in our search for relevant studies.

### 4.2 Eligibility

Table 2 specifies the inclusion and exclusion criteria used in the title and abstract screening process for this review.

  We included studies that described the impacts of HISs that were actually implemented and adopted for managing patient flow or solving patient flow problems. We excluded papers describing prototype systems, systems that were not implemented in practice or papers without real impacts of HISs on patient flow management such as those just reporting simulated results, simulation tests, or prediction models. Studies that focused on measures such as length of stay (LOS), and waiting time for clinical purposes without any relation to or discussion of patient flow management purposes were also excluded from this review.

**Table 1. Search terms.**

| Keyword | Boolean | Additional Keyword |
|---|---|---|
| *information system | AND | Patient flow or Hospital flow |
| Electronic health record or EHR | | Patient throughput |
| Electronic medical record or EMR | | Patient journey |
| Decision support system or DSS | | Overcrowding |
| Business intelligence system or BI system | | Access block |
| Computerised provider order entry system or CPOE | | Waiting time |
| Electronic bed board systems | | Length of stay |

First, we used the term "information system" (IS) as a general search term and the asterisk (*) because it could include different types of IS used in hospitals such as hospital IS, health IS, healthcare IS, or departmental ISs such as ED IS, intensive care unit (ICU) IS. Acknowledging the fact that hospitals adopt various types of IS with specific terminologies, we also included specific ISs commonly used by hospitals such as electronic health record (or EHR) or electronic medical record (or EMR), decision support systems (or DSS), business intelligence (BI) system, computerised order entry (CPOE) system and electronic bed board systems. In a similar vein, "patient flow" was used as a main keyword together with other synonyms such as "hospital flow", "patient throughput", "patient journey". Common indicators of patient flow management such as length of stay and waiting time were also included in the search. In addition, we used the bibliography of the selected papers to reach further studies. This technique is known as backward snowballing [15].

**Type of studies.** Apart from excluding simulation studies and review papers, we imposed no restrictions on the study's design or publication date as long as the studies examined the effects of HIS on patient flow management.

**Participants.** We included studies that were conducted in various healthcare settings including teaching hospitals, specialist hospitals and general hospitals (both public and private) and clinical centres. As long as the studies were conducted in these settings, we imposed no restrictions on the number of departments, units or wards involved. We also selected studies that addressed patient flow management at the network level, i.e., between different hospital sites and hospital centres. Studies investigating interventions in services not directly related to patient flow and patient access (such as financial services or insurance) were excluded from our review.

**Type of intervention.** Health information system is a broad concept and hospitals generally adopt and use several types of information systems to manage their operations. In this

**Table 2. Inclusion and exclusion criteria applied in the screening process.**

| | Inclusion | Exclusion |
|---|---|---|
| Study setting | • Study settings that include, primary, secondary, tertiary hospitals and clinical centres.<br>• Studies investigate the use of IS for patient flow management in hospital units such as ED or ICU were also included for review | • Studied outside these settings |
| Type of Interventions | • Computerised information systems used by hospitals or clinic centres | • Other interventions<br>• Single medical tools such as a CT scanner or heart measuring device.<br>• Personal Devices |
| Type of publication | • Articles published in peer-reviewed journals and conferences<br>• Full-text available<br>• English language | • Other types of publication such as: book chapters, reports, non-scholarly publications, reviews<br>• Full-text is not available<br>• Published in other languages |

review, we selected studies that addressed any type of computerised information systems that have been implemented and had impacts on patient flow outcomes. We also excluded paper-based information management systems, personal digital assistance devices, and medical tools such as surgery robots, CT scanners, heart rate measuring devices.

### 4.3 Study selection

To assist the selection of eligible studies for this review, we used the Preferred Reporting Items for Systematic Reviews and Meta-Analyses (PRISMA) guidelines with four key phases [16].

Initially, the first author searched through the pre-identified online databases by using combinations of the keywords to identify related studies. Duplicates were subsequently removed by using a tool called Covidence [17] and manually double checking by the first author. In the second step, two reviewers scanned the abstract of all studies to remove irrelevant or ineligible studies based on the predefined inclusion and exclusion criteria. The remaining studies went into the third step in which two reviewers assessed the full-text studies and further eliminated irrelevant papers. The final phase involved extracting data from included studies. We endeavoured to look for full-text files of the eligible papers in all resources available including using intra-library service to retrieve as many as possible

### 4.4 Data extraction and quality assessment

Information from the papers was extracted in the final list using an electronic data extraction form. Each study was given a unique identification number to ensure a consistent way of identifying studies between the two reviewers. The following data were extracted: authors, journals where the studies were published, year of publication, hospital's country, the study settings, study objectives, study design, description of the information systems used, factors affecting the adoption of HISs for patient flow management, the effects of HISs on patient flow outcomes, study results, study limitation and research gaps (S4 File, Example of data extraction form).

The GRADE [18] approach was adopted to assess the overall quality level of the evidence based on their design. GRADE approach provides particular useful guidelines for assessing health technology studies with heterogeneous study designs. Using the guidelines, the quality of evidence would be assessed as follows:

- High quality for randomized trial studies without serious limitations

- Low quality for observational studies

- '0' level of quality for studies where quality is not assessable such as expert opinion and studies without objective evidence.

### 4.5 Analytic frameworks

We adopted literature review and expert opinion to develop frameworks that describe types of HISs, their functional capabilities, and associated benefits (Table 3). We also used the conceptual model of Donabedian [19] as a framework for the analysis on patient flow outcomes. Donabedian's model categorises care quality into three groups: structure, process, and outcomes.

## 5. Results

The literature search returned 5996 studies and the removal of duplicates reduced the number to 5095. After the first level of screening in which we screened the titles and abstracts and applied the exclusion criteria, 4824 studies were removed. We then proceeded to screen the

**Table 3. Analytic frameworks of health information systems.**

| Framework | Reference | Elements |
|---|---|---|
| *Types of HIS* | Expert opinion | • Patient tracking system<br>• Electronic Health Record (EHR) systems<br>• Electronic Medical Record (EMR) systems<br>• Computerised provider order entry (CPOE) systems<br>• Dashboard systems<br>• Workflow management systems<br>• Clinical document management systems Other |
| *HIS capability* | Expert opinion | • Patient or event tracking<br>• Document management<br>• Order entry<br>• Patient registration<br>• Bed management<br>• Decision support<br>• Discharge management<br>• Patient flow reporting<br>• Prescription management Other |
| *Patient flow outcomes* | Donabedian model [19] | • Structure: measures related to healthcare centres' capability to deliver care such facilities, human resource.<br>• Process: measures related interactions between care providers and patients, and how care providers deliver care.<br>• Outcomes: measures related to the end results of the care processes. |

full-text of 271 studies and 231 of them were excluded. In addition, four studies were added to the final pool through the reverse snowballing technique. Details of the screening process is summarised in Fig 1, following the PRISMA flow diagram [16].

We included 44 studies for our systematic review. The included studies reported mixed impacts of HISs on patient flow management, which can be categorised as follows:

- 33 studies reported positive impacts [9, 20–51]

- 7 studies reported negative impacts [52–58] and 4 studies reported no impacts of ISs [59–62]. However, among the seven studies with negative impacts, two [55, 58] found that the negative effects were temporary and the patient flow measures returned back at pre-implementation baselines.

## 5.1 Types and features of the HIS

The included 44 studies reported the impacts of 17 different types of HIS on patient flow: eight EHR systems, eight EMR systems, seven patient tracking systems, four computerised provider order entry systems (CPOE), three patient flow dashboard systems, three departmental information systems including ED (1) and Radiology (2), and one each for workflow management, admission prediction, documentation management, patient scheduling, medical prescribing, patient discharge management, patient referral management system, bed management, consultation management, clinical information management, and Asthma management. Table 4 summarises details of the study site and publication profile of the included studies (publication year, country and study settings).

Research on the application of HISs to patient flow management can be dated back to the 1980s; however, it has gained prominence over the last decade. A majority of included studies were published in the period 2011–2020 (63.6%), compared to 29.5% of the 2001–2010 period and 6.8% of the 1988–2000 period. In addition, most of the studies selected for this review were published in developed countries where their governments have implemented promotional programs to increase the adoption of HISs in the healthcare sector. The number of

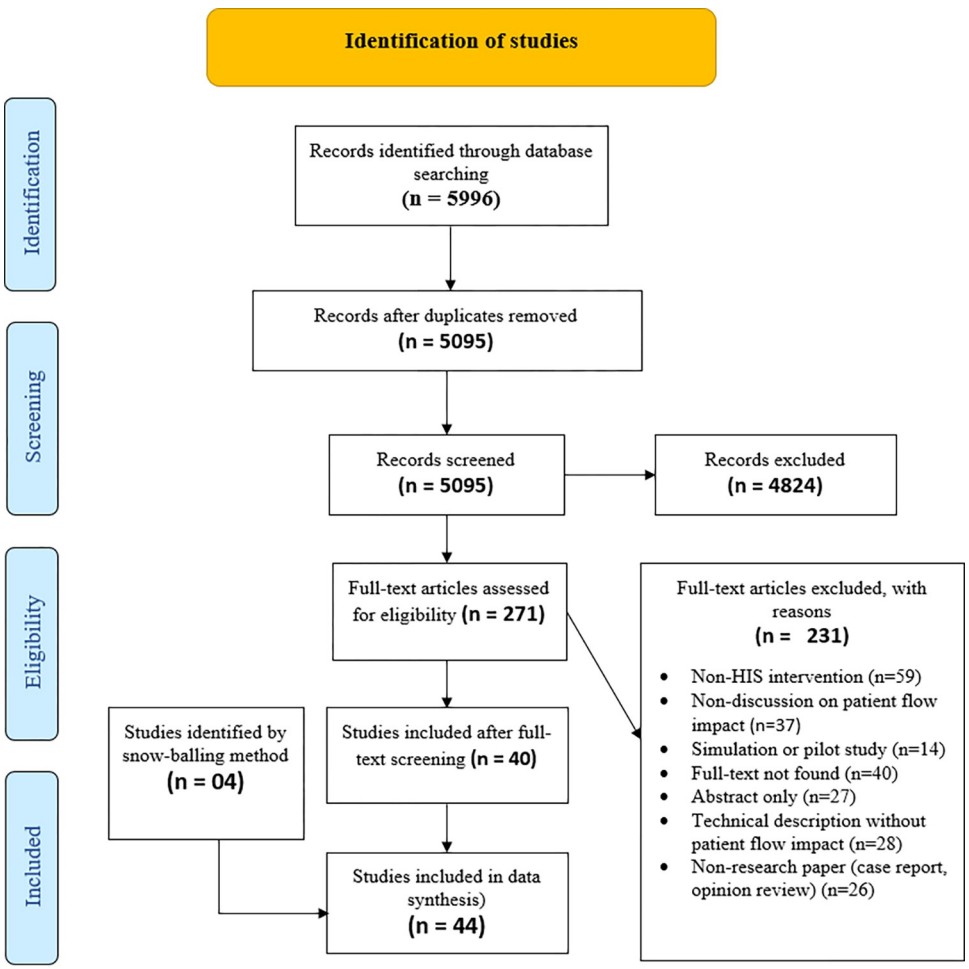

From: Moher D, Liberati A, Tetzlaff J, Altman DG, The PRISMA Group (2009). Preferred

Reporting Items for Systematic Reviews and Meta-Analyses: The PRISMA Statement. PLoS Med

6(7): e1000097. doi:10.1371/journal.pmed1000097

**Fig 1. Study screening process.** *From*: Moher D, Liberati A, Tetzlaff J, Altman DG, The PRISMA Group (2009). Preferred Reporting Items for Systematic Reviews and Meta-Analyses: The PRISMA Statement. PLoS Med 6(7): e1000097. doi:10.1371/journal.pmed1000097.

studies from the USA was the highest with 24 studies, followed by Australia with nine studies. Canada and South Korea contributed three and two studies, respectively. One study was conducted in each of the followings: England, Italy, Japan, Portugal, Uganda, and Taiwan.

In terms of settings, 20 of the reviewed studies discussed the impacts of HIS interventions at the department level, while eleven studies addressed hospital-wide level and three studies address network-wide level. Within the department level, 15 studies focused on EDs, three in Radiology and two in Paediatrics. Studies focused on hospital-wide patient flow when they include the coordination between several departments or units. For example, Westbrook et al. [51] discussed the impacts of CPOE on the flow of patients between ED and Pathology departments in Australian hospitals. In addition, we found that four studies described the impacts of HISs on patient flow across hospital networks [9, 31, 39, 42]. Fig 2 depicts where the reported HIS were studied in the care continuum and the number of studies.

**Table 4. Publication and study site profile of included studies.**

| Classification Criteria | Variables | Frequency (number of studies) | % |
|---|---|---|---|
| Publication year | 1988–2000 | 3 | 6.8% |
| | 2001–2010 | 13 | 29.5% |
| | 2011–2020 | 28 | 63.6% |
| Country | | | |
| | USA | 24 | 54.5% |
| | Australia | 9 | 20.5% |
| | Canada | 3 | 6.8% |
| | South Korea | 2 | 4.5% |
| | England, Italy, Japan, Portugal, Uganda, Taiwan (one study originated from each) | 6 | 13.6% |
| Study setting | | | |
| | Departmental level | 22 | 50% |
| | *Emergency Department* | *17* | *38%* |
| | *Radiology Department* | *3* | *7%* |
| | *Paediatric Department* | *2* | *5%* |
| | Hospital-wide | 18 | 41% |
| | Network-wide | 4 | 9% |

Specific functions of the 17 types of HIS that were described in the 44 studies included patient or event tracking (12 studies), clinical documentation management (12 studies), order entry (8 studies), patient registration (3 studies). Bed management, decision support, discharge management, patient flow reporting and prescription management were each included in three studies. Alert, disease detection, picture archiving, staff performance management, referral management, and reminder, were each discussed in one study. Almost all of the included HISs had the capability to integrate data from other systems. Twelve studies did not describe system features. Details of the HIS features and reported benefits are provided in S2 File.

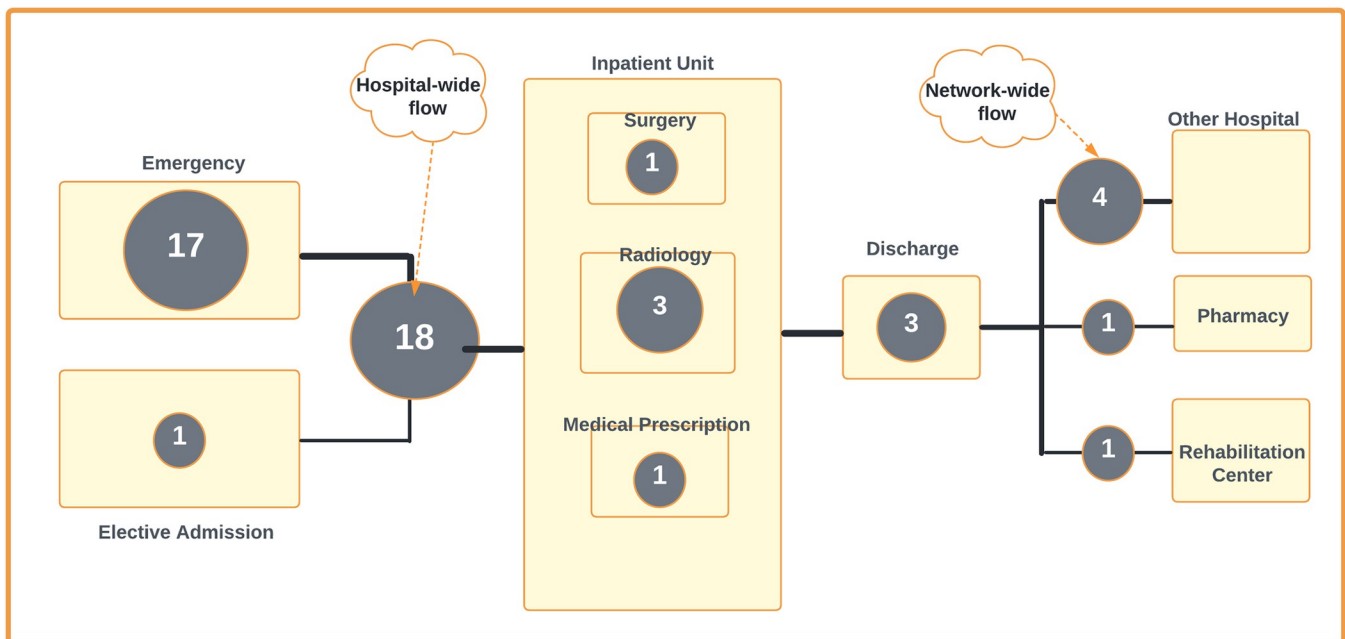

**Fig 2. Focused patient flow areas of the reported HISs.** The numbers in the circles correspond to the number of relevant studies reviewed.

## 5.2 Impacts on patient flow measures

Table 5 provides a summary of how key patient flow measures were grouped into three categories based on Donabedian model [19] and the number of studies that included these measures. Details of the included studies and HISs' impacts on patient flow measures are provided in S3 File, Characteristics of all included studies and their findings.

**Impacts on outcome measures.** Outcomes measures were the most studied measures in the included papers. This is not surprising because health outcomes are the end products of care and the target of health interventions. Studies examined two main types of outcome measures: individual outcomes and organisational outcomes. Almost all studies focused on individual outcome measures in which time-related measures included LOS (25), waiting time (13), treatment time (6), test turnaround time (TAT) (6), and boarding time (2). The effects of HISs on these time related measures were mixed. With regard to LOS, following the use of HISs: 14 studies [22–25, 29, 32, 33, 36, 38, 39, 41, 43, 45, 51] reported a decrease in LOS, 7 studies [9, 52–55, 57, 62] reported an increased LOS and 4 studies [58–61] found no difference. While most of these studies measured LOS in the ED, five studies [25, 39, 41, 43, 45] measured

**Table 5. Patient flow measures and the citation number of included studies.**

| Type of Measures | Detail | Citation Number |
|---|---|---|
| **Outcome Measure** | | |
| *Individual outcome* | | |
| | LOS | [9, 22–25, 29, 32, 33, 36, 38, 39, 41, 43, 45, 51–55, 57–62] |
| | Waiting time | [20–23, 31, 35, 42, 43, 46, 53, 54, 56, 58, 59] |
| | Treatment time | [22, 23, 44, 58–60] |
| | TAT | [21, 22, 38, 41, 48, 51] |
| | LWBS | [33, 54, 56, 61] |
| | Patient satisfaction | [35, 58, 62] |
| | Boarding Time | [22, 43, 52] |
| | Readmission rate | [37] |
| | Mortality | [32] |
| *Organisational outcome* | | |
| | Hospital costs | [33, 41, 47, 53] |
| | Film saving | [41] |
| | Staff satisfaction | [49] |
| | Staff stress level | [35] |
| **Process Outcome** | | |
| | Number of treatment or Medical staff productivity | [31, 41, 55], |
| | Number of early discharges | [37, 49] |
| | Guideline Adherence | [60] |
| | Number of shifts per staff | [56] |
| | Number of prescriptions | [49] |
| **Structure outcome** | | |
| | Room utilisation or occupation rate | [27] |
| | Number of diversions | [33] |
| | Average number or % of Access block | [27] |
| | EMS site avoidances | [38] |
| | ED patients with > = 12 LOS | [33] |
| | % of AV offload waiting time > 30m | [56] |

inpatient LOS and two studies [23, 59] reported changes in patient LOS at paediatrics centres. The ED LOS was not consistently defined. ED LOS was defined as the difference between ED exit time and the recorded arrival time [52]. Whereas some studies [22, 59] calculated detailed components which constitute the total LOS including time from arrival to triage, arrival to doctor, doctor to disposition, most other studies just reported the mean LOS.

Similarly, 14 studies reported impacts of HISs on waiting time. The results were mixed with 10 positive changes (reduction in waiting time), 2 negative and one with no statistically significant difference. Waiting time measures included waiting for the doctor, waiting for medical treatment, waiting for consultation and for examination. Three studies [43, 58, 59] examined impacts of EHRs on patients waiting for doctor time. In one study [59], investigators measured the mean patient flow time in a paediatric practice in the USA and found that although the mean patient flow time increased from 56.24 min to 81.43 min one month after the EHR implementation and to 64.60 min 12 months later, patients' waiting time (check-in to front desk and front desk to triage) actually dropped down by 1.51 and 9.33 min. Their findings suggested the EHR led to more positive results than negative because it reduced waiting for administrative works, allowing more time to be spent on treatment activities. Two studies [54, 56] reported negative impacts of HIS on the waiting time of ED patients. Gray and Fernandes [54] examined the adoption of CPOE in an ED in London Health Sciences Centre with around 100,000 patients per year to determine that CPOE caused an average increase of 5 min in waiting time. A more significant increase in waiting time from 40 to 78 min was observed in a 54,000 patient-per-year ED with the EMR system by Mohan, Bishop, and Mallows [56].

Treatment time is an important component of LOS and it directly influences health outcomes. However, in this review, we could only identify six studies that used this measure to assess the effectiveness of HIS. Unfortunately, these studies did not provide detailed explanation how the HIS affected treatment time. Three studies [22, 23, 44] found that health providers reduced treatment time when using an ED information system and a patient tracking system for their practice. The patient tracking system was used in a paediatric centre with 24,000 visits annually and it reduced the time of faculty paediatricians spent in Exam room from 11.33 to 6.53 min [23]. Meanwhile Baumlin et al. [22] determined a dramatic decrease by 1.90 h in the doctor-to-disposition time after an ED information system was implemented. Two studies about the EHR systems [58, 59] and an asthma management system [60] did not identify a significant difference in treatment caused by the interventions.

TAT is another time-related measure and it was investigated in 6 studies [21, 22, 38, 41, 48, 51]. TAT is defined as the time lapse between when the test is ordered and when the result is available [41]. Four studies [21, 22, 39, 41] examined TAT of the radiology examinations and laboratory results; one study [48] investigated TAT of housekeeping services and one study [51] reported pathology examinations. All of the studies reported impressive reductions in TAT after the implementation of a HIS. For example, Nitrosi et al. [41] noted a decrease in the mean chest exam TAT from 33.9 to 9.62 h.

Finally, boarding time is an important patient flow measure that is often referred to as access block or bed block and it is a main patient flow problem [63]. Two studies [22, 52] examined this measure although Pyron and Carter-Templeton [43] investigated provider discharge-to-nurse discharge time, which can be related to boarding time, but they did not explain or describe how this measure was calculated. Baumlin et al. [22] reported that the use of an ED information system reduced boarding time for the patient by 28% from 6.77 h to 4.90 h. By contrast, Feblowitz et al. [52] noticed an increase in the mean boarding time per patient from 211.2 min to 221.4 min in the long term (1 year after the implementation of an electronic documentation system) in an ED. However, neither study provided a causal relationship between HIS implementation and the changes in boarding time.

In addition to time-related measures, included studies also investigated other important individual outcomes including: four studies on the percentage of patients who left without being seen (LWBS), three studies on patient satisfaction, one each for mortality rate, and readmission rate. LWBS was studied in the ED setting. Three studies [54, 56, 61] reported increases of LWBS percentage with the most significant increase being reported in the study about a CPOE system from 24.3% to 42.0% [54], while Jensen [33] determined a reduction of 7.6%, but this study did not provide any subjective evidence. Patient satisfaction was measured in three studies with one positive result [35], one negative [62], and one neutral result [58]. The EHR system was found to reduce ED patient satisfaction because it increased LOS; however, the negative impacts lasted for only eight weeks before returning to the baseline from before the intervention implementation [62]. One study reported that the use of a patient discharging system [37] was associated with improvement in LOS for early discharge patients without higher rates of readmission. In another study, Inokuchi et al. [32] investigated the impacts of a newly-developed EMR system on the mortality rate at 28 days after hospitalisation and found no changes resulting from the intervention, which is a positive outcome.

Apart from the patient-related outcome measures above, studies also examined organisational outcomes including four studies about hospital costs, and one each for staff satisfaction, film saving and staff stress level. Three studies [33, 47, 53] calculated the reduction in LOS as hospital cost saving. The first study found that EMR systems were associated with 5.9% to 10.3% higher cost per discharge while with the implementation of a patient flow system, Jensen [33] reported that the hospital saved between 67,800 and 214,200 USD. The transition from traditional into digital radiology room through the implementation of a PACS system was found to reduce 90% of the film [41]. Staff satisfaction was examined in a study [49] which reported positive outcomes after the implementation of an electronic prescribing system. In a study about a workflow management system, Li et al. [35] found that the intervention greatly improved sonographers' productivity while reducing their stress level, which was measured by a 5-point Likert scale. Measures related to organisational outcome are an interesting part of the HIS literature because most of the evidence in patient flow intervention focused primarily on patient-related outcome measures.

**Impacts on process measures.**   Studies examined a variety of measures related to staff productivity in clinical processes, and medical guideline adherence. Four studies examined the effect of HIS on the number of medical services performed by the staff. Two studies showed increased number of surgeries [31] and radiology tests [41]. Nitrosi and colleagues [41] studied the impacts of a PACS and found that the number of imaging procedures increased by 7% although the number of technologies and radiologists remained unchanged. An increase of 37% in the number of surgeries after a surgery information system was observed by Gomes and Lapao [31]. However, EHR implementation was found to decrease the number of patients that clinical staff could see [55] although the negative impact was only temporary and resolved three months post-implementation. The implementation of HIS did not change the medical guideline adherence of the staff when they are already providing care that adheres to the relevant guideline [60]. The number of patients seen per shift by medical staff was measured by Mohan, Bishop, and Mallows [56] in an investigation of the effectiveness of an EMR system and the impact was negative. Mathews et al. [37] and Tran et al. [49] both measured the impact of HIS on the percentage of early discharged patients and show positive outcomes. Finally, Tran et al. [49] reported an increase in the number of prescriptions prepared the day before discharge as a positive effect of a prescription system.

**Impacts on structure measures.**   Evidence on the impact of HIS on structure measures was more limited than data on process and outcome measures. Six of the 44 studies reported some data on structure measures. These structure measures are related to flow problems facing

healthcare organisations and they were studied in ED settings. Almost all of the six studies reported positive impacts of HIS on these structure measures including the number of patient diversions and the number of ED patients with LOS over 12h [33], the proportion of early discharged patients [37], ED avoidance percentage [38], and the number and proportion of access blocks and hospital occupancy rates [27]. The study of Crilly et al. [27] found that the number of access blocks and hospital occupancy rates did not change after the implementation of a patient admission prediction system, but this is actually a positive outcome because the hospital presentations were increasing during the study period. By contrast, in one study, Mohan, Bishop, and Mallows [56] investigated the effect of an EMR system on the percentage of ambulance offloading time of more than 30 min which is also known as ambulance boarding and they found that the percentage went up from 10.5% to 13.3%.

### 5.3 Quality assessment of the included studies

Using the GRADE approach to assess the quality level of the evidence through their study design, two RCT studies [32, 60] were assessed as high quality and 38 observational studies using retrospective or prospective data were rated low quality. Four studies including three expert opinions and one stating improvement without figures did not provide objective evidence and they were rated '0' (the lowest rating). Two studies using multi-method design with both qualitative and quantitative components were rated low quality, based on the assessment of their quantitative component. Details of the quality assessment are provided in S5 File, Quality assessment of the included studies.

## 6. Discussion

### 6.1 Summary of key findings

This systematic review summarised and synthesised evidence from studies about HISs that have been applied to improve patient flow in both inpatient and outpatient settings. Overall, 33 out of the 44 included studies reported positive impacts of HIS on patient flow measures while 7 determined negative impacts, and 4 studies reported no significant impact. Half of the studies focused on patient flow at the departmental level; however, 18 studies reported the impact of HIS on the hospital-wide level and 4 studies reported network-wide impacts on HIS. Healthcare settings adopted at least 17 types of HIS to address patient flow problems and improve care efficiency.

We found that core features of the HIS interventions, that affected patient flow, included patient tracking, documentation management, order entry, patient registration, bed management, decision support, discharge management, prescription management and patient flow reporting. When it comes to the impacts of HIS on specific patient flow measures, most studies focused on outcome measures at both: patient (individual) and organisational level. Changes in individual outcomes were evident in time-related measures including length of stay (LOS), waiting time, treatment time, test turnaround time (TAT), and boarding time, and other measures such as left without being seen and patient satisfaction. Organisational outcome measures were noted in hospital costs, film saving, staff satisfaction, and staff stress level. Process measures and structure measures, although less examined in the included studies than outcome measures, are important measures. While process measures related to staff productivity and guideline adherence, structure measures included flow problems such as patient diversion, access block, hospital occupancy, ambulance offloading time, and ED patient with LOS over 12 h.

Noted HIS benefits included improvements in various patient flow aspects: access to needed information, staff communication, care coordination, work processes, and decision

support. Ineffective interaction between hospital units is one of the most common causes of poor patient flow [64]. HISs were effective in fostering care coordination and collaboration among multidisciplinary teams by imposing a common set of flow key performance indicators (KPIs), and metrics into practice. The application of these common, sometimes "simple", rules help develop common understandings and it is a key to governing complex systems [12]. In addition, the involvement of all team members in the development process of HIS is critical to achieving shared understandings. In this review, the effectiveness of HIS in care coordination was evident in many care processes such as patient check-in [59], elective waiting list management [31], bed management [36, 48], ambulance distribution [38], and discharge [36, 37, 49]. By integrating information from multiple siloed systems, patient flow-related HIS reduce the time needed for care providers to acquire sufficient information to make critical decisions. Real-time data, notifications, and alerts functions are key features that enabled users to get the most updated information in a timely manner. The development of HISs often included redesigning the embedded care process or processes, an opportunity for care settings to eliminate redundant steps and apply best practices to their care processes. Streamlined work processes helped reduce waiting time for test results and free up staff from redundant information [22, 34]. In addition, high degree of automation resulting from the HIS adoption contributed to the reduction in human errors, which can cause medical and health complications, and cognitive workload for hospital staff as they were not required to remember complex rules.

However, it still remains unclear how and why these interventions produced or did not produce positive or negative impacts. Most of the included studies were observational, before and after studies, making it challenging to establish the cause and effect link between HIS interventions and changes in patient flow measures. This has important implications because without a thorough understanding of why and how HIS affected patient flow, it is difficult to generalise the findings to other healthcare settings.

## 6.2 Strength and limitations

To date, several systematic reviews have been conducted to investigate interventions addressing patient flow problems; however, they focused mostly on operational methods such as triage, fast track, streaming [2]. Systematic reviews on the impact of HIS on patient flow are small in number and limited to single specific systems such as CPOE [5]. To the best of our knowledge, this review was the first attempt to evaluate a broad range of HISs applied in patient flow management. The novelty of this review lies in its research aim, and inclusion criteria, unlike most previous reviews on patient flow interventions, here, we included different types of HISs and broad scope of healthcare settings including departmental, organisational and network levels. Our findings provide different stakeholders with important insights for their implementation and adoption of HISs to optimise patient flow.

However, this review has several limitations. The first relates to the heterogeneous nature of the search terminology and the quantity and scope of the evidence. Although we conducted a comprehensive search, in many important domains, we could only identify a limited number of studies. The second limitation relates to the synthesis of varied outcomes and a broad range of HISs. In this review, we attempted to address this limitation by adopting analytic frameworks, which were based on domain experts and published literature, and by synthesising not only the health information system but also their functional features. Third, descriptions of the HIS interventions and the implementation process were often very limited, making it challenging to fully assess the system features and associated benefits. Fourth, most of the included studies are before-and-after, observational studies and therefore understanding of how and why HISs affected patient flow outcomes was very limited. Finally, we decided not to include a

meta-analysis because of the diverse, heterogeneous outcomes reported in the included studies. A meta-analysis, in this case, is inappropriate and can be more of a hindrance than a help [65].

## 6.3 Implications for patient flow management practice

Hospitals and care centres have implemented several interventions to tackle patient flow problems to deliver optimal care. However, up until recently, most of the efforts were focused on addressing ED overcrowding problems [3, 66, 67]. It is evident in the literature that focusing solely on ED problems will not likely achieve optimal flow because EDs do not operate separately, rather they are part of an interconnected system [68]. Therefore, literature has urged that patient flow needs to be viewed from the whole system of care viewpoint and called for a shift from ED-focused to system-wide or hospital-wide interventions [12, 69]. However, the gap between understanding the problem and having solutions to solve the problem seems still far. For example, even a holistic approach like Lean healthcare was still attached to a specific department or care process [4]. The frequently reported intervention to improve inpatient flow was implementing a specialised staff or team to coordinate patient flow across hospital units; however, the solution still posed significant challenges [3]. This systematic review found that apart from 22 studies focusing on department level, many studies reported hospital-wide or even network-wide level. HISs' potential to address patient flow at the hospital-wide level were noted in their ability to improve communication between multidisciplinary teams [25, 36], enhance care coordination [36, 49], improve access to needed information [41, 43], and streamline care processes [25, 59]. One of the prominent causes of admission bottleneck is inefficient discharges [68] because any delays in inpatient discharge will increase hospital occupancy and ED overcrowding [69]. HISs showed their effectiveness in discharge prediction and established standardised discharge criteria for improving the discharge process [37]. These "medical-readiness criteria" have been shown to facilitate efficient planning and care coordination [37]. Addressing patient flow problems sometimes goes beyond hospital scope to a higher level of network-wide scope. A dashboard system was developed in Alberta, Canada to address ED overcrowding by coordinating emergency services between different emergency rooms within the region [9]. HISs were also used within a network of different hospitals to address the need for rehabilitation care services and improve the consultation process [42]. HISs can be scalable to a nationwide level to reduce waiting time for elective surgical patients [31]. By providing information about capacity, occupancy and demand, they can be highly effective in addressing the mismatch between supply and demand to improve patient flow.

## 6.4 Implications for future research

Moving forward, this review suggests important areas for future research in the field. First, additional studies need to explore barriers and facilitators of the HISs related to patient flow management. This will offer valuable implications for healthcare organisations to drive their HIS project to success and derive the most from their investment. Second, learning about the effectiveness of HISs on patient flow and associated factor during the post-implementation phase could help to advance the field. This is because of the evolutionary nature of HIS development in which factors associated with the application of HIS can be captured and used as lessons learned for the next evolution of the HIS [70]. In this review, only the study of Inokuchi et al. [32] addressed this topic. Patient flow is often negatively affected during the implementation of HIS because of changes in the workflow and human resources. Although the effect seemed temporary, learning about these periods and associated factors will bring implications for researchers and policymakers when considering the project timeline and expected

challenges. Furthermore, although HISs are found to help healthcare organisations address patient flow management areas such as care coordination, timely access to information, and communication barriers, understanding why and how HIS could enhance each of these aspects can be extremely helpful. Part of the reasons to explain this is because most of the selected studies in this review did not include adequate details of the underlining technologies of the HIS interventions such as: what are the technical supports and architectures, what are the input and output data, or how the output data are represented in the user interface. The lack of technical specifications of the HIS interventions made it hard to fully comprehend how they contributed to the changes in patient flow management. Finally, during the last two years, the COVID-19 pandemic has completely disrupted patient flow management all over the world. Yet, we could not identify any studies on the role of HIS in remedying the impacts of the pandemic on patient flow.

## 7. Conclusion

Health information systems (HISs) provide clear benefits in managing patient flow over traditional paper record management systems. However, without a systematic evaluation and summary of the available evidence, stakeholders interested in adopting HISs in healthcare settings for patient low management might be lost in the ocean of information. This is especially true when it comes to the questions of what HIS to invest, what benefits and impacts to expect and how to maximize the values from their investment. This systematic review has revealed an increasing interest in adopting HIS to address patient flow issues in healthcare settings in the last decade. HISs can be effective solutions for patient flow management at the organisational-wide or even network-wide levels due to their great scalability and integrability. HISs were often found to be effective in improving communication and care coordination between team members, providing timely access to high quality information for decision making, and streamlining care processes. These improvements contributed to more efficient patient flow throughout the care continuum. As more healthcare and health-related data are generated, there are great opportunities for HISs such as decision support systems, and dashboard systems to help healthcare organisations harness the power of big-data analytics and achieve optimal patient flow. This review shows that HISs can impact various aspects of patient flow at different levels of care; however, how and why they delivered the impacts will require further research.

## Supporting information

**S1 File. Search strategy.**
(DOCX)

**S2 File. Reported benefits of HISs on patient flow management.**
(DOCX)

**S3 File. Table of all included studies and findings.**
(XLSX)

**S4 File. Example of data extraction form.**
(XLSX)

**S5 File. Quality assessment of the included studies.**
(DOCX)

**S6 File. PRISMA checklist.**
(DOC)

## Acknowledgments

We would like to thank the Faculty of Information Technology (Monash University) subject librarian, Mario Sos for his great expertise and valuable feedback in developing the search strategy. We are grateful for generous help from Quang H Vo in screening the titles, abstracts and full-text papers in our review. Also, Angela Melder from Monash Centre for Health Research and Implementation, Monash University and Monash Health gave us valuable feedback on the inclusion/exclusion criteria during the screening process.

## Author Contributions

**Conceptualization:** Quy Nguyen, Michael Wybrow, Frada Burstein.

**Data curation:** Quy Nguyen.

**Formal analysis:** Quy Nguyen.

**Resources:** Quy Nguyen.

**Supervision:** Michael Wybrow, Frada Burstein, David Taylor, Joanne Enticott.

**Writing – original draft:** Quy Nguyen.

**Writing – review & editing:** Quy Nguyen, Michael Wybrow, Frada Burstein, David Taylor, Joanne Enticott.

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
