## [Decision Letter · Decision Letter 0]

22 Jun 2022

PONE-D-22-14339Understanding the impacts of health information systems on patient flow management: A systematic review across several decades of researchPLOS ONE

Dear Dr. Nguyen,

Thank you for submitting your manuscript to PLOS ONE. After careful consideration, we feel that it has merit but does not fully meet PLOS ONE’s publication criteria as it currently stands. Therefore, we invite you to submit a revised version of the manuscript that addresses the points raised during the review process.

We look forward to receiving your revised manuscript.

Kind regards,

Yong-Hong Kuo

Academic Editor

PLOS ONE

Journal Requirements:

"This systematic review is part of a PhD research fully funded by a hospital and a university."

"QDN was supported by a Ph.D. scholarship jointly funded by the Monash University Graduate Research Industry Partnership (GRIP) program and by Eastern Health. The Funders of this work did not have any direct role in the design of the study, its execution, analyses, interpretation of the data, or decision to submit results for publication."

"No authors have competing interests"

5.In your Data Availability statement, you have not specified where the minimal data set underlying the results described in your manuscript can be found. PLOS defines a study's minimal data set as the underlying data used to reach the conclusions drawn in the manuscript and any additional data required to replicate the reported study findings in their entirety. All PLOS journals require that the minimal data set be made fully available. For more information about our data policy, please see http://journals.plos.org/plosone/s/data-availability.

Additional Editor Comments:

The manuscript has been reviewed by two referees. Both have found that the work is meaningful and worth publication. They have provided some comments and suggestions to the authors for the revision.

Reviewers' comments:

Reviewer's Responses to Questions

**Comments to the Author**

1. Is the manuscript technically sound, and do the data support the conclusions?

Reviewer #1: Yes

Reviewer #2: Partly

2. Has the statistical analysis been performed appropriately and rigorously? 

Reviewer #1: Yes

Reviewer #2: I Don't Know

3. Have the authors made all data underlying the findings in their manuscript fully available?

Reviewer #1: Yes

Reviewer #2: Yes

4. Is the manuscript presented in an intelligible fashion and written in standard English?

Reviewer #1: Yes

Reviewer #2: Yes

5. Review Comments to the Author

Reviewer #1: This is a meaningful review paper. The structure is well-organized. Some parts need to be improved:

Minor points:

1. In the "4. Method", the sub-titles seem to have typos. The titles should be "4. X", rather than "1. X".

2. The "6. Discussion" contains much information. Please make it more concise and use some sub-titles or sub-sections to construct this section in a more logical manner.

Major points:

1. The authors listed some previous HIS papers' practical effects without these HISs' underpinning technologies. These HISs' technical supports should also be considered in this review.

Such that, which company / what database technologies provide this specific HIS service. What does the UI present / How does this HIS work in the investigated papers.

2. Adding more figures and tables will make this review more logical and clearer.

For example, the "5. Results" section presented the reviews with various sub-sections. Each sub-section can have a corresponding table, which has columns like, "paper title", "year", "HIS type", "impact", etc.

3. This review paper is a "systematic review across several decades of research". Therefore, it could be better if the systematic logic of this review paper can be highlighted. Why this review is systematic? What sub-systems or specific sub-domains does this review study. What are the changes in HISs during these decades? What are the HISs' differences among different countries?

4. The authors also mentioned the influences of the COVID-19 pandemic. Did COVID-19 affect the selected HIS papers in this review? Are these HISs before COVID-19, or after COVID-19?

5. Please present the review through more figures, tables, and various pictures, rather than pure text. The figures can make this review more straightforward, and deliver more information.

Reviewer #2: Appreciating your work, kindly find hereunder some comments.

1. Can the authors please review the PLOS guidelines for manuscript submission and revise the contents of the manuscript accordingly. Particularly in question are the "objective, research question" segments. Additionally, the lack of line numbers make it difficult to give comments.

2. There is significant overlap in content included in the results and discussion sections. Suggest revision.

6. PLOS authors have the option to publish the peer review history of their article (what does this mean?). If published, this will include your full peer review and any attached files.

Reviewer #1: No

Reviewer #2: No

---

## [Author Response · Author response to Decision Letter 0]

30 Jul 2022

II. Reviewer 1’s comments

2.1 In the "4. Method", the sub-titles seem to have typos. The titles should be "4. X", rather than "1. X".

• We corrected the sub-sections to 4.1, 4.2, etc. as suggested by the reviewer.

2.2 The "6. Discussion" contains much information. Please make it more concise and use some sub-titles or sub-sections to construct this section in a more logical manner.

• We added the following sub-titles to reflect the content in “6. Discussion” to divide it into logical themes.

6.1 Summary of the key findings

6.2 Strength and limitations

6.3 Implications for patient flow management practice

6.4 Implications for future research

• We removed some of the text deemed redundant or repetitive in this section.

2.3 The authors listed some previous HIS papers' practical effects without these HISs' underpinning technologies. These HISs' technical supports should also be considered in this review.

Such that, which company / what database technologies provide this specific HIS service. What does the UI present / How does this HIS work in the investigated papers.

• We highly appreciate this feedback and believe that this recommendation is very relevant to make our review more interesting and valuable. In response to this, we added a supporting table in S3_File that lists all of the information system interventions of the included studies. We also re-visited all of the included studies in our review and tried to extract the technical aspects of the included information system interventions. Unfortunately, many of the included studies in our review did not contain technical details of the interventions. The few studies that contain some technical descriptions make it infeasible to tabulate the database technologies, UI presentation, HISs’ underpinning technologies or how the HIS work in a meaningful way. We completely agree that this lack of technical descriptions limits our understanding of the impacts of these HIS interventions on patient flow management, and this is a limitation of the included studies. We acknowledged this limitation in Section 6 and recommended future research to overcome this limitation.

2.4 Adding more figures and tables will make this review more logical and clearer.

For example, the "5. Results" section presented the reviews with various sub-sections. Each sub-section can have a corresponding table, which has columns like, "paper title", "year", "HIS type", "impact", etc.

• We added a new table (Table 5) to the Section 5 to provide an overview of our result section.

• We also believe that our Fig.2 is an interesting visualisation of the impact of HIS on different areas of the whole care continuum.

2.5 This review paper is a "systematic review across several decades of research". Therefore, it could be better if the systematic logic of this review paper can be highlighted. Why this review is systematic? What sub-systems or specific sub-domains does this review study. What are the changes in HISs during these decades? What are the HISs' differences among different countries?

• We added a text under Table 4 to describe changes in the application of HIS in patient flow management practice over time and the differences between countries. We believe the addition of this text together with Table 4 will provide the audience with more complete information.

• In regard to the question “Why this review is systematic?”, we believe this question is related to our method of conducting this systematic review. If this is the case, we provided a detailed description of our method in Section 4. In this section, we specified how we searched for publications including a systematic search strategy, how we selected relevant studies, extracted and analysed the information including an analytical framework which was used for our review.

2.6 The authors also mentioned the influences of the COVID-19 pandemic. Did COVID-19 affect the selected HIS papers in this review? Are these HISs before COVID-19, or after COVID-19?

• We reviewed our discussion on this point and can confirm that:

o The HIS interventions in the included studies were before COVID_19.

o The reason why we mentioned the pandemic is because it may be an additional complicating factor that may be needed to be considered in future research.

2.7 Please present the review through more figures, tables, and various pictures, rather than pure text. The figures can make this review more straightforward, and deliver more information.

• We believe this comment and comment 2.4 are similar and so we added a new table to Section 5.

III. Reviewer 2’s comments

3.1 Can the authors please review the PLOS guidelines for manuscript submission and revise the contents of the manuscript accordingly. Particularly in question are the "objective, research question" segments. Additionally, the lack of line numbers makes it difficult to give comments.

• We have added line numbers to the current version of the manuscript.

• We have re-visited PLOS One’s guidelines for formatting and revised our manuscript accordingly.

• We added introductory text to Section 3 to relate our research questions to the whole review.

3.2 There is significant overlap in content included in the results and discussion sections. Suggest revision.

• We found that there were major repetitions in the text in Section 5 and Section 6 of our review. Therefore, we removed these major sections as follows:

o Section 6.1: We removed the discussion on the type of HIS interventions because the information was described in Section 5.

o Section 6.1: We removed the discussion on the patient flow measures and the Donabedian model because it repeated Section 5.2

---

## [Decision Letter · Decision Letter 1]

30 Aug 2022

Understanding the impacts of health information systems on patient flow management: A systematic review across several decades of research

PONE-D-22-14339R1

Dear Dr. Nguyen,

We’re pleased to inform you that your manuscript has been judged scientifically suitable for publication and will be formally accepted for publication once it meets all outstanding technical requirements.

Kind regards,

Yong-Hong Kuo

Academic Editor

PLOS ONE

Additional Editor Comments (optional):

Based on the referees' recommendations, I recommend Accept.

Reviewers' comments:

Reviewer's Responses to Questions

**Comments to the Author**

1. If the authors have adequately addressed your comments raised in a previous round of review and you feel that this manuscript is now acceptable for publication, you may indicate that here to bypass the “Comments to the Author” section, enter your conflict of interest statement in the “Confidential to Editor” section, and submit your "Accept" recommendation.

Reviewer #1: All comments have been addressed

Reviewer #2: All comments have been addressed

2. Is the manuscript technically sound, and do the data support the conclusions?

Reviewer #1: (No Response)

Reviewer #2: Yes

3. Has the statistical analysis been performed appropriately and rigorously? 

Reviewer #1: (No Response)

Reviewer #2: Yes

4. Have the authors made all data underlying the findings in their manuscript fully available?

Reviewer #1: (No Response)

Reviewer #2: Yes

5. Is the manuscript presented in an intelligible fashion and written in standard English?

Reviewer #1: (No Response)

Reviewer #2: Yes

6. Review Comments to the Author

Reviewer #1: (No Response)

Reviewer #2: I would like to thank the authors for taking the time and employing a much needed effort in bettering the contents of their manuscript as per the suggested revisions.

7. PLOS authors have the option to publish the peer review history of their article (what does this mean?). If published, this will include your full peer review and any attached files.

Reviewer #1: No

Reviewer #2: No

---

## [Editor Report · Acceptance letter]

2 Sep 2022

PONE-D-22-14339R1 

*Understanding the impacts of health information systems on patient flow management: A systematic review across several decades of research*

Dear Dr. Nguyen:

I'm pleased to inform you that your manuscript has been deemed suitable for publication in PLOS ONE. Congratulations! Your manuscript is now with our production department. 

Kind regards, 

on behalf of

Dr. Yong-Hong Kuo 

Academic Editor

PLOS ONE